# Associations between Maternal Education and Child Nutrition and Oral Health in an Indigenous Population in Ecuador

**DOI:** 10.3390/ijerph20010473

**Published:** 2022-12-28

**Authors:** Bharathi Chinnakotla, Sita Manasa Susarla, Deepika Chandra Mohan, Bathsheba Turton, Hannah M. Husby, Cecilia Paz Morales, Karen Sokal-Gutierrez

**Affiliations:** 1School of Public Health, University of California, Berkeley, CA 94704, USA; 2Henry M. Goldman School of Dental Medicine, Boston University, Boston, MA 02118, USA; 3General Dental Practice, Quito 170513, Ecuador

**Keywords:** oral health, public health, maternal-child health, social determinants of health, maternal education, nutrition, early childhood caries, ecuador

## Abstract

The global nutrition transition has increased the prevalence of childhood dental caries. Greater understanding is needed of the impact of social determinants—including maternal education—on child oral health. This is a cross-sectional analysis of a convenience sample of families of 458 indigenous Ecuadorian children aged 6 months through 6 years from 2011–2013. Data was collected by mother interviews and child dental and anthropometric examinations. Multivariate logistic and Zero-Inflated-Poisson regression analyses assessed associations between years of maternal education and maternal-child oral health practices and child oral health outcomes. Each additional year of maternal education was significantly (*p* < 0.05) associated with some healthier practices including greater likelihood of mothers and children drinking milk daily (OR 1.20; 95% CI 1.08, 1.34); and less healthy practices including greater likelihood of bottle-feeding children with sugary liquids (OR 1.14; 95% CI 1.06, 1.22) and to older age, giving children sweets daily, calming children with a bottle or sweets, and less likelihood of helping brush their children’s teeth (OR 0.93; 95% CI 0.88, 0.98). Each year of maternal education had a small but statistically non-significant influence on increasing the odds of children being among those who are cavity-free (OR 1.03; 95% CI 0.92, 1.16). Interventions to improve health outcomes should focus not just on maternal education but also address social and commercial determinants of health through nutrition and oral health education, as well as policies to reduce sugar and ensure universal access to oral health care.

## 1. Introduction

Over recent decades, low- and middle-income countries (LMICs) have undergone a nutrition transition from breastfeeding to bottle-feeding and from traditional diets to ultra-processed and sugary foods and beverages. These dietary shifts have contributed to dramatic increases in the prevalence of diet-related chronic diseases including dental caries, obesity, type 2 diabetes, and cardiovascular disease [1,2].

Dental caries is the most prevalent chronic disease of childhood, affecting 60–90% of children worldwide [2]. Early childhood caries (ECC), defined as tooth decay in children under six years of age, can cause severe dental infections, mouth pain, and interfere with a child’s nutrition, growth, development, and overall health and quality of life [3,4]. ECC is a multifactorial disease, and the interplay of risk and protective factors has been illustrated through different models. The Keyes Caries Triad delineates the interaction between dietary sugar, cariogenic oral biofilm, and underlying child health factors contributing to caries [5]. The Caries Balance Model depicts the opposition of caries-causing factors (e.g., dietary sugars) and caries-preventive factors (e.g., fluoride and saliva) [6].

Many studies have shown that children’s oral health outcomes are strongly affected by social and structural factors. A conceptual model of determinants of oral health included social factors at family and community levels (e.g., parent education, economic factors, cultural practices, access to foods/beverages and dental care) that can contribute to or prevent caries [7]. Recent models of the Social Determinants of Oral Health also incorporate commercial determinants of oral health, including the global marketing of sugary drinks and ultra-processed foods that drive the nutrition transition, as well as political, economic and research priorities, and racism, which can adversely affect access to services and health outcomes for racial and ethnic minority populations [8,9,10]. In LMICs and low-income populations in high-income countries, cross-sectional studies have demonstrated that prolonged bottle-feeding and frequent consumption of sugary drinks and snacks contribute to increased risk of ECC [11]. However, systematic reviews, and cross-sectional and longitudinal studies of social determinants of health (e.g., maternal education and family income) have shown contrasting associations with ECC—some positive, some negative, and some “U-shaped” with higher caries rates in both low- and high-income populations [12,13,14,15,16,17].

Multiple cross-sectional studies have indicated that individuals with higher socioeconomic status (SES) tend to have healthier practices and better health outcomes than those with lower SES [18,19,20]. Specifically, much evidence points to education as playing a key role, as education can improve one’s knowledge, habits, skills, and resources, which in turn improve one’s health behaviors and lifestyle choices, leading to overall healthier outcomes [20,21]. Therefore, education level is often considered a good proxy measure of SES in oral health studies and maybe a focal point for intervention.

Ecuador is a middle-income Latin American country that has experienced a nutrition transition with persistently high rates of malnutrition and increasing rates of obesity and ECC [22,23,24,25,26]. Ecuador currently has free and compulsory public education from age 6 to 14, including primary school for grades 1 through 6 [27], and free public healthcare, including dental care; however, low-income, rural and indigenous populations have experienced barriers in access to education and healthcare. The most recent oral health data from the 2010 national health survey found that 80% of 6-year-old children had dental caries [24]. There are wide economic and health disparities, especially in rural and indigenous populations who have higher rates of poverty and stunting malnutrition, as well as high rates of dental caries [23,25,28,29,30,31,32,33,34,35]. A study in the Amazonian region of Ecuador found that 92% of indigenous and 95% of non-indigenous 6-year-old children had dental caries, with a mean of 6.4 and 8.4 decayed teeth, respectively [30].

Global health organizations have advocated policies and community-based interventions to promote oral health, especially for underserved and indigenous populations [36,37,38,39]. However, the social determinants of ECC in Ecuador, particularly for rural and indigenous communities, remain understudied, and very few studies have specifically examined the relationship between maternal education and child nutrition or oral health-related behaviors and outcomes [28,29,30,31,32,33,34,35]. This study aimed to explore whether mothers in a rural, indigenous population in Ecuador who had a greater number of years of education might also have children with better oral health outcomes. 

## 2. Materials and Methods

### 2.1. Study Design and Population

This is a cross-sectional study of the relationship between years of maternal education and child oral health in a convenience sample of 458 children aged 6 months through 6 years and their mothers/caregivers living in Pueblo Kichwa de Rukullakta, a network of 17 rural, indigenous communities in Ecuador’s Amazonian Napo Province. This analysis examines first visits (baseline) between 2011 and 2013 for families participating in Alli Kiru, a community-based program promoting early childhood nutrition and oral health (Figure 1). The program was a collaboration among University of California Berkeley, local leaders of the Kichwa community, Ecuadorian Ministries of Health and Education, and Rotary International. The study was approved by the University of California, Berkeley Committee for the Protection of Human Subjects (#2011-04-3178), with approval by local partner leadership; and performed in compliance with the Helsinki Declaration.

### 2.2. Data Collection and Measures

Kichwa community volunteers and child care/preschool teachers invited all families in their communities with children aged 6 months through 6 years to participate in the program. Trained volunteers fluent in Spanish and/or Kichwa obtained mother/caregiver informed consent and child assent and interviewed mothers/caregivers in their preferred language. The questionnaire was adapted for a low-literacy population from the World Health Organization (WHO) Oral Health Survey [40] with 50 questions addressing family demographic characteristics, oral health and nutritional knowledge and practices, and assessment of their child’s oral health status. Previously published studies describe the details of participant recruitment [41], as well as the details of utilizing the questionnaire in Ecuador and other low- and middle-income countries [41,42,43,44]. Child dental examinations were conducted by 7 licensed dental examiners (dentists and dental hygienist) from Ecuador and US, with a trained assistant, by visual inspection with a light and dental mirror [40]. They recorded decayed, missing, and filled teeth (dmft), and estimation of depth of cavitation by inspection: into enamel only, into the dentin, and deep cavitation close to or in the pulp. The examiners did brief standardization independently and then jointly examined 5 children and agreed on the findings. Children were weighed and measured (by length or height), without shoes and in lightweight clothes, by trained volunteers using a digital scale and stadiometer (Seca, Chino, CA, USA) [45].

Community volunteers were provided a small stipend, oral health education materials in Kichwa and Spanish, and toothbrushes and toothpaste for their families. Preschool and childcare teachers were given toothbrushes and toothpaste holders for their classrooms and toothbrushes and toothpaste for their families. Participating families were provided nutrition and oral health education in Kichwa and/or Spanish, toothbrushes and toothpaste for all family members, fluoride varnish applications to children’s teeth, child dental screening and onsite dental treatment or referrals for treatment as needed, and enrollment in a raffle of children’s toys and clothes.

### 2.3. Statistical Analysis

The analysis was guided by this study’s hypothesis that years of maternal education is associated with maternal-child nutrition and oral health practices and child oral health outcomes. We developed a Directed Acyclic Graph or DAG to select potential confounders and covariates and to identify complex relationships between maternal education and oral health practices and outcomes (Figure 2). This analysis utilized maternal education level as a proxy measure associated with SES in LMICs, since the questionnaire did not directly inquire about family income due to cultural sensitivity concerns and the predominance of subsistence farming rather than income in this community. Many maternal-child health studies have included parental education in measurements of socioeconomic status and demonstrated that maternal education is a good predictor of child health outcomes and can serve as a proxy for SES [46,47,48,49,50].

Data cleaning was completed in Excel and Google Sheets (Figure 1), and analyses were completed in RStudio (version 4.0.3, RStudio PBC, Boston, MA, USA), Microsoft Excel (version 2201, Microsoft Corporation, Albuquerque, NM, USA), and Google Sheets. The questionnaire and exam data included continuous and categorical variables. Some categorical variables were binary while others consisted of multiple categories that were later converted to binary categories.

Child nutrition status was calculated from child height, weight, sex, and age based on the WHO growth reference standards for children [37]. WHO AnthroPlus software version 3.2.2 for children under age 5 and WHO AnthroPlus version 1.0.4 for children aged 5 and above (Geneva, Switzerland) was used to generate Z scores for Height-for-Age (HAZ), Weight-for-Age (WAZ) and BMI-for-Age (BAZ). Malnutrition was categorized as stunting (HAZ < −2), underweight (WAZ < −2) and wasting (BAZ < −2). BMI-for-age was defined as overweight (Z > +2) and as obese (Z > +3) for children under age 5, and overweight (Z > +1) and obese (Z > +2) for children aged 5 and above [45,51].

In our univariate analysis, we first calculated descriptive statistics to identify patterns in maternal-child oral health and nutrition practices as well as child oral health status outcomes. We then performed logistic regression analyses to explore the relationship between years of maternal education and mother-child oral health practices, adjusting for the confounders and covariates selected based on the literature on social determinants of oral health [11,12,13,14,15] (Figure 2). For continuous variables, a correlation test was performed to obtain the Pearson’s correlation coefficient, adjusting for the aforementioned confounders and covariates. Zero-Inflated Poisson (ZIP) regression was performed to assess the relationship between years of maternal education and child oral health outcomes (dmft). The logistic component of the ZIP model was utilized to assess the odds of decay for all children and the Poisson component of the ZIP model was utilized to predict the counts for children with decay.

## 3. Results

### 3.1. Family Demographic Characteristics

The demographic characteristics of participating mothers, children, and families are shown in Table 1. A total of 252 mothers and 458 children aged 6 months through 6 years were studied. Mothers had a mean age of 30 years, approximately 6 years of education, and an average of 4 children. Children’s mean age was 3.7 years and a slightly larger proportion were female (56.4%) than male (43.6%). Households had a mean of 7 people, with several indicators of low socioeconomic status including 17% cooking with wood only, 14% lacking electricity, and 72% lacking potable water. Approximately two-thirds of families (68%) lived less than a 5 minutes’ walk to the nearest store that sold sugary drinks and ultra-processed snacks (i.e., junk food).

### 3.2. Maternal-Child Oral Health Practices

Mothers’ and children’s oral health practices are shown in Table 2. Most mothers had received prenatal care (79.3%), but less than two-thirds (64.5%) had ever visited a dentist. A small proportion of mothers drank milk daily (20%) and very few drank soda daily (6%). Roughly half of mothers (52%) reported currently suffering from an oral health problem such as dental pain, decayed teeth, or bleeding gums. Mothers’ knowledge of child oral health was limited. Only one-third (34%) knew 2 or more causes of caries (e.g., dietary sugar and poor oral hygiene), and less than one-third (29%) knew 2 or more consequences of caries (e.g., mouth pain, difficulty eating and sleeping, poor overall health).

Most children (80%) were reported to be up-to-date on vaccinations, but less than one-third of children (28%) had ever been to the dentist. Regarding nutrition practices, nearly all children were breastfed (97%) for a mean duration of 17 months. One-third of children were bottle fed (34%) for a mean duration of 13 months. One in five children (18%) were given sugary liquids in the bottle, and 4% slept with the bottle. One in five children (20%) drank milk daily, and over half of children (52%) consumed junk food daily–including 6% who drank soda, 14% who ate sweets, 10% who ate chips, and 21% who ate sugary ice pops. When mothers were asked how they calmed their child when fussing, 27% reported giving their child food, 6% the baby bottle, and 7% sweets. Regarding oral health practices, over three-quarters of children (77%) were reported to have their own toothbrush and over 70% had toothpaste, but only one-third (36%) of mothers helped their child brush. Child malnutrition was categorized as stunting (28.1%), wasting (1.1%), underweight (7.2%), and overweight or obese (4.2%).

### 3.3. Child Oral Health Outcomes

Child dental exams showed that more than three-quarters of children (77%) had tooth decay with a mean dmft of 6.7. Most decay (87%) was untreated, with active decay in a mean of 5.9 teeth, 0.4 missing teeth due to caries, and 0.6 filled teeth. Nearly 4 in 10 children (39%) currently complained of mouth pain. Mothers were more than twice as likely to report that their child had poor oral health (13%) compared to poor overall health (6%) (Table 3).

### 3.4. Association between Maternal Education, Maternal-Child Nutrition and Oral Health Practices, and Child Oral Health Outcomes

The associations between years of maternal education and maternal-child nutrition and oral health practices are shown in Table 4, with maternal education used as a continuous predictor for all regression analyses. After adjusting for confounders and covariates, a one unit increase in the years of maternal education was significantly associated with some greater health knowledge and healthier practices—greater likelihood of mothers knowing at least 2 consequences of child caries (Adjusted Odds Ratio (aOR) = 1.17), mothers drinking milk daily (aOR = 1.20), and giving their children milk daily (aOR = 1.20). However, increase in years of maternal education was also significantly associated with some less healthy child nutrition and oral health practices—greater likelihood of bottle-feeding the child (aOR = 1.10), feeding sugary liquids in the bottle (aOR = 1.14), calming a fussy child with the bottle (aOR = 1.16), bottle-feeding to an older age (*r* = 0.20), giving the child sweets daily (aOR = 1.11), calming a fussy child with sweets (aOR = 1.16); and less likelihood of helping brush the child’s teeth (aOR = 0.93).

The results of the ZIP regression models between years of maternal education and child oral health outcomes are shown in Table 5. Among children with decayed teeth, a one-unit increase in the years of maternal education was found to significantly decrease the expected number of child total dmft by a factor of 0.99, and decrease the expected number of untreated decayed teeth by a factor of 0.98, while holding all other variables constant. Among all children, a one-unit increase in the years of maternal education was found to increase the odds of a child being in the zero dmft group (i.e., to be cavity-free) by a factor of 1.03, increase the odds of a child being among those with zero untreated decayed teeth by a factor of 1.04, increase the odds of a child being among those with zero missing teeth due to caries by a factor of 1.03; and decrease the odds of a child being among those with zero filled teeth by a factor of 0.94 (i.e., increase the odds of having fillings for decayed teeth), although the findings were not statistically significant.

## 4. Discussion

This cross-sectional study of a convenience sample of children aged 6 months through 6 years and their mothers/caregivers from a rural indigenous Ecuadorian community explored the associations between maternal education and maternal-child oral health practices and child oral health outcomes. Specifically, we identified 2 pathways: the education-practice relationship to understand the relationship between years of maternal education and oral health practices, and the education-outcome relationship to understand the relationship between years of maternal education and child oral health outcomes (primary study outcome).

In the education-practice relationship, we found that a one-unit increment in the years of maternal education was significantly associated with some healthy practices as well as many unhealthy practices, but not significantly associated with the mother’s knowledge of the causes of childhood caries, mother or child dental visits, prenatal care, child vaccination, and child nutritional status. In the education-outcome relationship, a one-unit increase in the years of maternal education was shown to significantly decrease the expected number of a child’s total dmft and the expected number of a child’s untreated decayed teeth.

This study supports the findings of other systematic reviews, meta-analyses, and cross-sectional studies showing high rates of ECC in Ecuador, including in indigenous populations [26,30,52,53]. Our results are consistent with findings from other cross-sectional studies in LMICs regarding the widespread practice of feeding young children sugary beverages and snacks [54,55], limited parental knowledge about the contribution of the baby bottle and sugary beverages to ECC [56], and the need to help children with tooth brushing until age 8 [57]. Our study also supports other global studies showing limited access to dental care, especially for low-income, rural, and indigenous populations, and widespread suffering of children and adults from untreated dental caries [2,3,8].

This study contributes to the global debate on the association between maternal education and child oral health, for which systematic reviews, cross-sectional and longitudinal studies have shown disparate findings [12,13,14,15,16]. Our study suggests that years of maternal education was not predictive of oral health knowledge, but it did predict some healthier and unhealthier practices, which may tip the balance toward either greater risk or lower risk for ECC depending on the local social and commercial determinants of health. We hypothesize that, in this population, maternal education may contribute to greater likelihood of employment outside of the home, which may lead to reliance on bottle-feeding, less time to brush their children’s teeth, added income, and desire to give their children treats that are accessible at their local store – including milk, soda, and junk food. A longitudinal study in Australia has shown that children of full-time employed mothers were more likely to consume high-sugar drinks compared to children of part-time employed mothers [58]. A recent global study found that women’s economic empowerment was associated with many benefits, but it was also associated with greater risk for ECC [59].

Our findings also suggest that academic schooling alone is unlikely to improve oral health knowledge and oral health and nutrition practices and outcomes. Most schools do not include effective instruction on healthy diets, toothbrushing, and dental care [60]. To the contrary, children attending school are frequently exposed to many risks for poor nutrition and oral health, such as stores adjacent to the schools selling sugary drinks and junk food, mobile vendors coming to school sites to sell junk food, and daily school-sponsored snacks and beverages with high sugar content [61]. While most child oral health interventions rely on providing maternal education on oral health and dental treatment for children, it appears that the role of maternal education in determining child oral health outcomes may not be as strong as other socio-structural and commercial factors. Our study suggests a more complex set of social determinants of oral health, and a need for more intensive interventions at child, family, and community levels to address the socio-structural and commercial drivers of disease including expanded nutrition and oral health education, access to dental care for all children and adults, and implementing national and local policies to support basic income, nutrition and education, and limits on the adverse commercial determinants of oral health.

In 2014, the Pan American Health Organization and all Latin American countries signed an agreement to support school nutrition programs and prohibit non-nutritious snacks and beverages from schools, with the aim of preventing childhood obesity [62]. From 2014–2015, Ecuador implemented several important nutrition programs: (1) “Traffic-light” labeling of pre-packaged processed foods and beverages, indicating whether the product has high (red), moderate (yellow) or low (green) levels of sugar, fat, and salt; (2) Regulations on school food vendors prohibiting sale of products high in sugar, salt, or fat and requiring sale of fruits and vegetables, and provision of free, safe water; and (3) An added tax on processed sugary drinks of USD 0.18 per 100 g of sugar. Evaluations of these regulations found moderate levels of compliance and improvement in nutrition awareness and emphasized the need for ‘scaling up’, monitoring and enforcement of the regulations, and increased funding for health promotion [56,57]. Another evaluation found that families continued to call for greater access to healthy foods and limits on marketing of unhealthy products to their children, while food industry representatives and some government officials continued to advocate for industry ‘rights’ to market their products without regulation, and academics called for further research [59].

While Ecuador has focused on policies to promote nutrition, there is an unmet need for equal focus on oral health. The World Health Organization, FDI World Dental Federation, and the Lancet Commission on Oral Health have advocated a global life-course approach to improving oral health, including oral health education and dental clinical services incorporated into primary health care as part of Universal Health Care from prenatal care through well-child/immunization programs and adult healthcare, utilizing a broader primary health care network including nurses and community health workers, and incorporating toothbrushing programs into childcare and schools [8,60,61,62,63,64,65,66,67,68]. Experts also emphasize that engaging the local community in the design and implementation of services, as well as dismantling structural racism, are critical to ensure accessibility, acceptability, and effectiveness [63].

This study contributes to the literature by elucidating the relationships between maternal education and maternal-child oral health practices and oral health outcomes in this LMIC population. This study has many limitations: The cross-sectional nature of the study does not allow for temporality to be established, and therefore we cannot prove causality. The convenience sampling method may limit the generalizability of the findings beyond this specific population, but also serves as a strength for the inclusion of diverse communities. Survey responses may be susceptible to recall bias and social desirability bias. There also may have been clustering at the family and community level for which this exploratory analysis did not account. While we conducted an exploratory analysis of the relation between years of maternal education and our mediator (oral health practices) and our primary outcome independently, it may be valuable to conduct a mediation analysis in the future. This analysis focused on only maternal education as a social determinant of oral health; but since we found that the role of maternal education was not as strong as prior studies have suggested, it may be worthwhile to explore the impact of other social, economic, environmental, and commercial factors on maternal-child oral health practices and outcomes. Finally, this study assessed community nutrition and oral health prior to the implementation of national policy interventions to improve child and adult nutrition. Future research should follow-up with these communities to assess the impact of the policies on nutrition and oral health and identify additional interventions that may be impactful.

## 5. Conclusions

This study of a rural, indigenous Ecuadorian community found low maternal education levels and high prevalence and severity of ECC. Increases in maternal education level were associated with some healthy and some unhealthy dietary and oral health practices; and overall, weakly associated with better ECC outcomes. There is a need to explore the impact of the broad socio-structural factors on oral health practices and outcomes, and to address the social and commercial determinants of ECC with interventions in childcare/schools and maternal-child health programs to incorporate nutrition and oral health education focused on promoting breastfeeding, limiting bottle-feeding and sweets, and brushing children’s teeth daily. Moreover, government health programs should be expanded to ensure access to dental treatment for children and adults; and regulations should be strengthened to protect children from the adverse health impacts of commercial interests.

## Figures and Tables

**Figure 1 ijerph-20-00473-f001:**
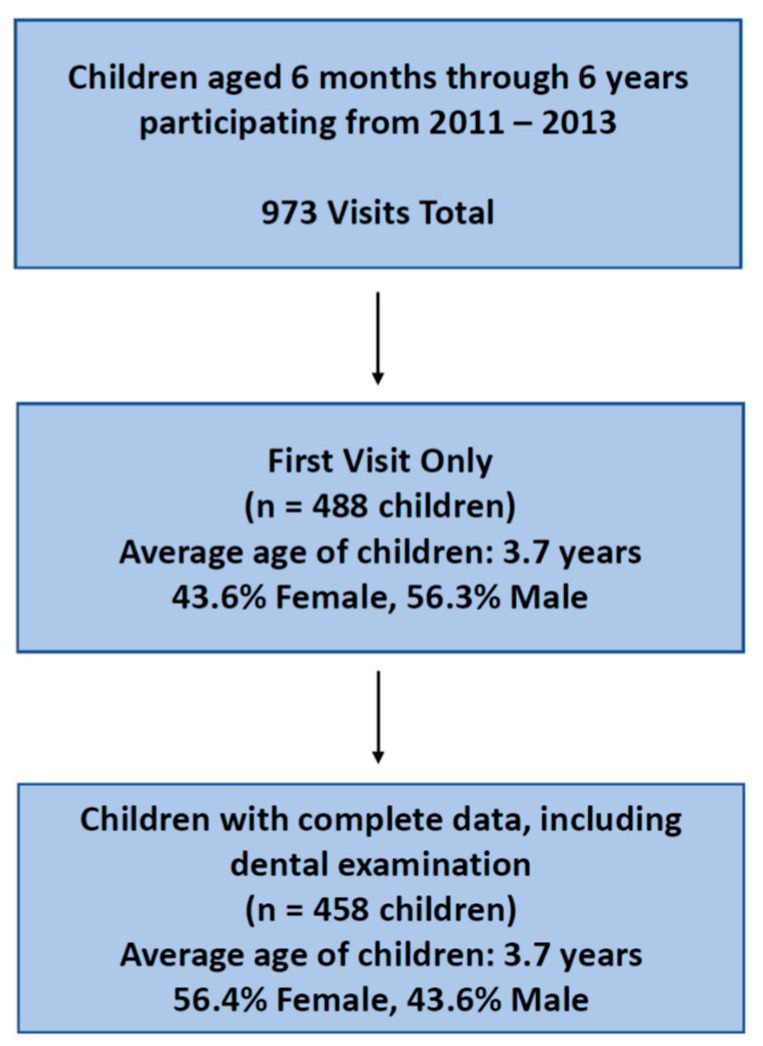
Analytic Sample.

**Figure 2 ijerph-20-00473-f002:**
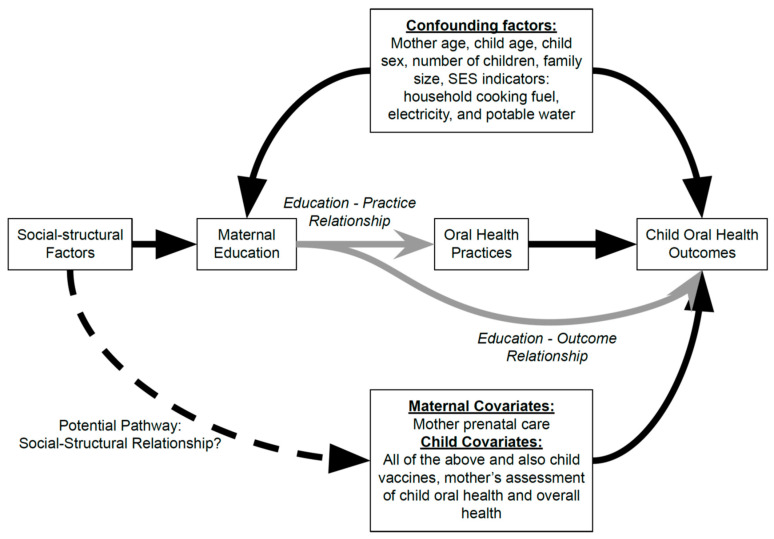
Hypothesized Relationship between Maternal Education, Maternal-Child Oral Health and Nutrition Practices, and Child Oral Health Outcomes. *Gray:* Our pathways of interest; *Dotted*: A potential, unexplored pathway.

**Table 1 ijerph-20-00473-t001:** Family Demographic Characteristics.

Characteristics	Mean ± SD or % *n* = 252 Mothers *n* = 458 Children
**Mother**	
Age (years)	30.0 ± 7.8
Education Level (years)	5.9 ± 4.2
Number of Children	4.0 ± 2.5
**Child**	
Age (years)	3.7 ± 2.5
Sex (%)	
Female	56.4
Male	43.6
**Family**	
Number of People in a Household	7.2 ± 2.8
Cook with Wood Only (%)	17.0
No Electricity at Home (%)	14.0
No Potable Water at Home (%)	72.0
Walking Distance to Store Selling Junk Food (%)	
<5 min	67.7
6–20 min	21.0
>20 min	11.3

**Table 2 ijerph-20-00473-t002:** Maternal-Child Oral Health Practices, and Nutrition.

Practices	Mean ± SD or % *n* = 252 Mothers *n* = 458 Children
**Mother**	
Received Prenatal Care (%)	79.3
Daily Dietary Consumption (%)	
Milk	19.6
Soda	6.3
Has Own Toothbrush (%)	91.7
Has Been to the Dentist (%)	64.5
Current Oral Health Complaints (e.g., dental pain, decayed teeth, bleeding gums, inflammation) (%)	52.1
Knows at Least 2 Causes of Child Caries (%)	34.2
Knows at Least 2 Consequences of Child Caries (%)	29.2
**Child**	
Vaccines Up-to-date (%)	80.2
Child Breastfed (including breast milk in bottle) (%)	97.2
Breastfeeding Duration (months)	17.0 ± 8.1
Child Bottle-Fed (%)	34.4
Bottle-Feeding Duration (months)	13.4 ± 10.8
Drank Sugary Liquid in Baby Bottle (%)	18.3
Slept with Baby Bottle (%)	4.1
Daily Dietary Consumption (%)	
Milk	19.6
Soda	6.3
Sweets	14.4
Chips	10.4
Ice Pops	21.3
Children Consuming Any Junk Food Daily (%)	52.3
How Mother Calms Fussing Child (%)	
With Food	27.2
With Baby Bottle	5.5
With Sweets	7.0
Has Own Toothbrush (%)	76.7
Has Own Toothpaste (%)	70.5
Mother Helps Child Brush Teeth (%)	35.7
Child Has Been to the Dentist (%)	28.5
Average Number of Dental Visits	1.7 ± 2.0
** *Nutrition Status* **	
Malnutrition (%)	
Stunting ^1^	28.1
Wasting ^2^	1.1
Underweight ^3^	7.2
Overweight or Obese ^4^	4.2

^1^ Stunting: Height-for-age Z < −2; ^2^ Wasting: BMI-for-age Z < −2; ^3^ Underweight: Weight-for-age Z < −2; ^4^ Overweight or Obese: BMI-for-age Z > +2 for children under age 5 and Z > +1 for children aged 5 and above.

**Table 3 ijerph-20-00473-t003:** Child Oral Health Outcomes.

Health Status	Mean ± SD or % *n* = 458 Children
** *Oral Exam* **	
Children with Tooth Decay (%)	76.7
Average Number of Decayed, Missing, or Filled Teeth (dmft)	6.7 ± 5.5
Untreated Decayed Teeth (d/dmft) (%)	87.3
Average Number of Decayed Teeth	5.9 ± 5.1
Average Number of Missing (Extracted) Teeth	0.4 ± 1.4
Average Number of Filled Teeth	0.6 ± 1.4
** *Mother’s Perception of Child’s Oral Health* **	
Current Child Complaints of Mouth Pain (%)	38.9
Mother Believes Child’s Oral Health is Poor (%)	12.6
Mother Believes Child’s Overall Health is Poor (%)	5.6

**Table 4 ijerph-20-00473-t004:** Relationship between Years of Maternal Education and Maternal-Child Oral Health Practices.

Practices	Unadjusted Odds Ratios (95% CI) or Pearson’s Correlation Coefficient (95% CI)	Adjusted Odds Ratios ^0^ (95% CI) or Pearson’s Correlation Coefficient (95% CI)	*p*-Value for Adjusted Estimate
**Mother**			
Received Prenatal Care ^1^	1.10 (1.01–1.20)	1.09 (0.99–1.20)	0.085
Drinks Milk Daily ^2^	1.19 (1.09–1.30)	1.20 (1.08–1.34)	<0.01 **
Drinks Soda Daily ^2^	0.99 (0.90–1.09)	0.95 (0.85–1.07)	0.41
Has Own Toothbrush ^1^	1.10 (0.99–1.23)	1.02 (0.89–1.16)	0.81
Has Been to the Dentist ^2^	1.04 (0.99–1.11)	1.03 (0.96–1.10)	0.39
Current Oral Health Complaints ^2^	1.00 (0.94–1.05)	1.01 (0.95–1.08)	0.74
Knows at Least 2 Causes of Child Caries ^2^	0.94 (0.88–0.99)	0.94 (0.88–1.00)	0.081
Knows at Least 2 Consequences of Child Caries ^2^	1.11 (1.04–1.18)	1.17 (1.08–1.27)	<0.01 **
**Child**			
Vaccines Up-to-date ^1^	1.05 (0.99–1.11)	1.07 (1.00–1.15)	0.067
Child Breastfed ^1^	0.95 (0.83–1.09)	0.85 (0.65–1.06)	0.17
Breastfeeding Duration ^3,5^	−0.0043 (−0.096–0.087)	−0.0039 (−0.032−0.024)	0.78
Child Bottle-Fed ^1^	1.10 (1.05–1.16)	1.10 (1.05–1.17)	<0.01 **
Bottle-Feeding Duration ^3,5^	0.20 (0.11–0.29)	0.20 (0.18–0.23)	<0.01 **
Drank Sugary Liquid in Baby Bottle ^2^	1.14 (1.08–1.21)	1.14 (1.06–1.22)	<0.01 **
Slept with Baby Bottle ^1^	1.09 (0.98–1.22)	0.98 (0.86–1.12)	0.80
Daily Dietary Consumption:			
Milk ^2^	1.18 (1.11–1.25)	1.20 (1.11–1.29)	<0.01 **
Soda ^2^	1.03 (0.94–1.21)	1.01 (0.91–1.12)	0.89
Sweets ^2^	1.10 (1.04–1.18)	1.11 (1.03–1.20)	<0.01 **
Chips ^2^	1.04 (0.97–1.11)	1.07 (0.98–1.16)	0.11
Ice Pops ^2^	1.03 (0.98–1.09)	1.05 (0.99–1.12)	0.13
How Mother Calms Fussing Child:			
With Food ^1^	0.99 (0.95–1.04)	1.04 (0.98–1.10)	0.17
With Baby Bottle ^1^	1.10 (1.00–1.21)	1.16 (1.03–1.31)	0.013 *
With Sweets ^1^	1.08 (0.99–1.17)	1.16 (1.04–1.29)	<0.01 **
Has Own Toothbrush ^1^	1.05 (0.99–1.11)	1.01 (0.93–1.09)	0.80
Has Own Toothpaste ^1^	1.08 (1.03–1.14)	1.03 (0.97–1.10)	0.33
Mother Helps Child Brush Teeth ^1^	1.00 (0.96–1.04)	0.93 (0.88–0.98)	0.011 *
Child Has Been to the Dentist ^1^	0.98 (0.93–1.02)	0.95 (0.89–1.01)	0.083
Number of Dental Visits ^3,5^	0.023 (−0.069–0.11)	0.023 (−0.0044−0.051)	0.099
** *Nutrition Status* ^4^ **			
Chronic Stunting Malnutrition	0.99 (0.94–1.04)	1.02 (0.96–1.08)	0.60
Wasting	1.06 (0.85–1.31)	1.48 (1.01–2.72)	0.10
Underweight	1.00 (0.91–1.08)	1.05 (0.95–1.16)	0.39
Overweight/Obese under age 5	1.02 (0.93–1.12)	1.01 (0.90–1.13)	0.88
Overweight/Obese age 5 and above	1.01 (0.84–1.19)	0.98 (0.78–1.23)	0.85
** *Mother’s Perception of Child’s Oral Health* **			
Current Child Complaints of Mouth Pain ^2^	1.00 (0.94–1.06)	0.99 (0.92–1.07)	0.82
Mother Believes Child’s Oral Health is Poor ^2^	0.96 (0.89–1.03)	— ^6^	— ^6^
Mother Believes Child’s Overall Health is Poor ^2^	0.84 (0.74–0.94)	0.90 (0.79–1.02)	0.10

* *p* < 0.05; ** *p* < 0.01; CI, Confidence Interval; ^0^ Mother Characteristics adjusted for mother age, number of children, family size, household cooking fuel, electricity, potable water, and mother prenatal care; Child Characteristics adjusted for child age, child vaccines, mother’s assessment of child oral and overall health, and child sex in addition to the covariates included for Mother Characteristics; ^1^ Binary variables where the null values or absence of the variable in question is the reference category; ^2^ Multi-categorical variables converted into binary variables; ^3^ Continuous variables; ^4^ All nutrition status variables converted into binary variables based on Z scores: Stunting: Height-for-age Z < −2; Wasting: BMI-for-age Z < −2; Underweight: Weight-for-age Z < −2; Overweight or Obese: BMI-for-age Z > +2 for children under age 5 and Z > +1 for children aged 5 and above; ^5^ Pearson’s Correlation Coefficient; ^6^ Model failed to converge.

**Table 5 ijerph-20-00473-t005:** Zero-Inflated Poisson Model: Relationship between Years of Maternal Education and Child Oral Health Outcomes.

Oral Health Outcomes	Count Model ^1^ Estimate (95% CI)	Count Model *p*-Value	Inflation Model ^2^ Estimate (95% CI)	Inflation Model *p*-Value
*Oral Exam*				
Number of Decayed, Missing, or Filled Teeth (dmft)	0.99 (0.98–1.00)	0.023 *	1.03 (0.92–1.16)	0.43
Number of Decayed Teeth	0.98 (0.97–1.00)	<0.01 **	1.04 (0.93–1.16)	0.37
Number of Teeth Missing Due to Caries	1.00 (0.92–1.08)	0.88	1.03 (0.91–1.20)	0.60
Number of Filled Teeth	1.03 (0.97–1.08)	0.19	0.94 (0.85–1.04)	0.13

^1^ Poisson with log link; for children with tooth decay; ^2^ Binomial with logit link; for all children in the sample; * *p* < 0.05; ** *p* < 0.01; CI, Percentile-Based Confidence Interval.

## Data Availability

The original datasets analyzed in this study are not publicly available in accordance with participant privacy, informed consent forms that did not include release of the data, and the study’s approved IRB protocols. However, the minimal dataset necessary to interpret, replicate and build upon the findings of this study are available from the senior author [KSG] upon reasonable request.

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
