# Peer review of "Associations between Maternal Education and Child Nutrition and Oral Health in an Indigenous Population in Ecuador"

_ijerph, 2022, doi:10.3390/ijerph20010473_

Round 1

Reviewer 1 Report

Clearly written. Figure 2 particularly helpful in describing rationale for this paper.  Figure 1 - unclear how 973 are included when only 488 children screened.  Perhaps some comparisons from interviews only, while others include oral health status -But demographic tables include n=458.  Results  tables should include N so that this is apparant.  

Table 5 - when reporting odds of filled teeth - is this among all children or only those with decay - same question for reported dental visits.

Background should include some description of the dental services system and the educational system in Ecudor.  What does 6 years of maternal education consist of?  Are dental visits uncommon in all of Ecudor, or more so in indigeonous community?

Author Response

Thank you for taking the time to review our manuscript. Please see the attachment for our response.

Reviewer 2 Report

Dear authors, 

I read with interest your scientific research for which I want to congratulate you.

I would suggest introducing in section keywords the term Early Childhood Carries as you take this disease into discussion a lot in your research.

Author Response

(The authors gave the same response as above.)

Reviewer 3 Report

·     The title is unclear. Suggest “Associations between maternal education and oral health-related nutritional status in Indigenous Ecuadorian children”.

·         Abstract: Suggest to include (1) the date of sampling took place; (2) the OR and p-value about the significance of results. In line 17, please clarify how many years?

·         Keywords: Please delete “public health”, and add “nutritional status”, “Ecuador”.

·         Line 44,53,55,59,70,283,285,288,293: Please define the type of studies (cross-sectional, longitudinal).

·        Line 59-64: What is the current knowledge gap? It is unclear to me why the study focused on maternal educational level only? What about other SES like maternal work status? What is the level of support mother participation in the labor force with regard to the context of the study? Mothers in the labor force are more likely to have children attending childcare/daycare centers. Participation by mothers in the labor force may lead to unhealthy eating habits among children. I would suggest authors referring to this interesting study (International Journal of Consumer studies 2018; 42(5): 522-532).

·         Line 79-81: Please revise the aim based on the title.

·      Line 84-95: Please describe in much more details how participants were recruited? What were the inclusion and exclusion criteria for participation in this study?

·       Line 101-104: Was the questionnaire pretested among participants? The use of a questionnaire should also be explained in greater depth, as well as justifying their use.

·         Line 105: How many US dentists or a dental hygienist?

·        Line 110-112: Did the authors consider assessing whether oral health associated with BMI? I would think this would be something useful to examine and I would suggest that you look into it and include the findings in your analysis.

·         Line 135-138: Please define these variables.

·         Figure 1 should be revised and should include the inclusion and exclusion criteria.

·         Please define the statistical analysis program used?

·         Line 261-282: Please do not repeat results here.    

·         Ref # 45 is very old- please delete/update

·         Authors should follow the journal guideline for referencing.

Author Response

(The authors gave the same response as above.)

Round 2

Reviewer 3 Report

Dear Authors,

Abstract: It would be benefit to include the main significant (OR, p value). It is not just say maternal education was significantly associated with some healthier practices.....etc.

Introduction: Good to refer to this article in the discussion. I understand that The dataset included maternal education level only, but why this study is important? Why maternal education level matter in Ecuador? What this study adds? I think the novelty of this study could be improved (Line 79-86).

Author Response

Comment 1: Abstract: It would be benefit to include the main significant (OR, p value). It is not just say maternal education was significantly associated with some healthier practices.....etc.

Response 1: Thank you for your comment. We have added the OR values and the associated 95% CI to the abstract. 

Comment 2: Introduction: Good to refer to this article in the discussion. I understand that The dataset included maternal education level only, but why this study is important? Why maternal education level matter in Ecuador? What this study adds? I think the novelty of this study could be improved (Line 79-86).

Response 2: We have modified our introduction to include the purpose and importance of this study. Please see lines 67-68 and 86-88.